# Risk factors influencing *Coxiella burnetii* seropositivity in water Buffalo (*Bubalus bubalis*) populations of Egypt's Nile Delta

Abdelfattah Selim[1]*, Mohamed Marzok[2]*, Hattan S. Gattan[3,4], Hesham Ismail[5]

1 Department of Animal Medicine (Infectious Diseases), Faculty of Veterinary Medicine, Benha University, Toukh, Egypt, 2 Department of Clinical Sciences, College of Veterinary Medicine, King Faisal University, Al-Ahsa, Saudi Arabia, 3 Department of Medical Laboratory Sciences, Faculty of Applied Medical Sciences, King Abdulaziz University, Jeddah, Saudi Arabia, 4 Special Infectious Agents Unit, King Fahad Medical Research center, King AbdulAziz University, Jeddah, Saudi Arabia, 5 Department of Public Health, College of Veterinary Medicine, King Faisal University, KSA

* Abdelfattah.selim@fvtm.bu.edu.eg (AS); mmarzok@kfu.edu.sa (MM)

## Abstract

This study investigated the seroprevalence of *Coxiella burnetii* in buffaloes across several governorates in Egypt's Nile Delta. No significant variation was observed between regions (P > 0.05), although the highest prevalence was recorded in Kafr El-Sheikh (15.5%) and the lowest in Menofia (9.1%). While sex was not statistically significant, females showed a higher seroprevalence (12.4%) than males (9%). Age had a significant impact, with buffaloes over 8 years of age showed a higher prevalence (21.6%) compared to the younger age groups. Tick infestation was also significantly associated with infection, with a prevalence of 22.4% in infested animals. Buffaloes exposed to communal grazing (13.8%) or kept in contact with small ruminants (16.3%) showed increased seropositivity. Notably, animals with a history of abortion had a markedly higher prevalence (26.7%). Multivariate logistic regression analysis identified age above 8 years (OR = 6.7), tick infestation (OR = 3.0), contact with small ruminants (OR = 3.0), and abortion history (OR = 3.2) as significant risk factors. Communal grazing (OR = 1.9) and age between 5 and 8 years (OR = 2.2) were also associated with increased odds of seropositivity. These findings highlight key epidemiological factors contributing to *C. burnetii* infection risk in buffaloes.

## 1. Introduction

*Coxiella burnetii*, a globally distributed zoonotic pathogen, is the causative agent of Q fever in both humans and animals [1]. This Gram-negative intracellular bacterium is present worldwide, with the exception of New Zealand and Antarctica [2–4].

 *C. burnetii* exhibits a complex life cycle and can infect a wide range of host species. Livestock, particularly cattle, goats, and sheep—serve as the principal reservoirs, shedding the bacterium through milk, vaginal secretions, feces, and birthing

**Data availability statement:** All data are in the manuscript.

**Funding:** This work was supported through the Annual Funding track by the Deanship of Scientific Research, Vice Presidency for Graduate Studies and Scientific Research, King Faisal University, Saudi Arabia (Grant KFU253068 to MM).

**Competing interests:** The authors have declared that no competing interests exist.

materials [5–7]. *C. burnetii* is primarily transmitted to humans through exposure to placental tissues and birth fluids following parturition [8]. Infected farms can also contaminate agricultural materials such as manure, hay, and straw, leading to the release of infectious aerosols into the environment [9,10].

In domestic animals, *C. burnetii* infection can cause abortion, stillbirths, and various reproductive problems, including anoestrus, repeat breeding, and chronic endometritis [8]. However, clinical signs in cattle are often absent or inconsistent with the infection. Infected cows may shed *C. burnetii* in their milk for up to 13 months without showing any visible symptoms, making the condition easy to overlook and frequently underdiagnosed [7,11]. *C. burnetii* can be transmitted through various routes, with inhalation being the primary mode of infection in both animals and humans [12]. The bacterium is highly resilient and can be dispersed over long distances by wind, especially when shed into the environment, facilitating its spread across open landscapes [13,14].

Ticks play a role in the transmission of *C. burnetii*, especially within sylvatic (wildlife) ecosystems. They can transmit the pathogen to subsequent generation through transovarial transmission, making them self-sustaining reservoirs of *C. burnetii* [15,16].

Mechanical vectors like flies, cockroaches, and lice can facilitate the spread of *C. burnetii*, while rodents serve as important reservoirs of the pathogen [17]. Additionally, domestic pets such as cats and dogs may contribute to urban outbreaks through aerosol transmission, particularly during abnormal birthing events [18,19].

To develop effective strategies for the management and prevention of Q fever, it is essential to identify potential risk factors, given the disease's multiple transmission routes and broad host range [20]. The detection of *C. burnetii* in humans and domestic ruminants underscores its wide distribution in Egypt [21–23]. The study of risk factors associated with *C. burnetii* infection in bovines in Egypt remains limited, particularly in buffaloes, which represent an important component of the country's livestock population. While cattle have received more attention in previous epidemiological surveys, data on the prevalence, and contributing risk factors in buffaloes are notably scarce [24]. Addressing this knowledge gap is essential for developing targeted surveillance, control, and prevention strategies for Q fever in Egypt's diverse livestock systems.

Therefore, the present study aims to determine the prevalence of *C. burnetii* infection in water buffaloes (*Bubalus bubalis*) and to assess the associated risk factors contributing to its transmission.

## 2. Materials and methods

### 2.1. Ethics approval

The study was approved by ethical committee of the faculty of veterinary medicine, Benha university. All procedures and methods were conducted according to guidelines of ethical committee of faculty of veterinary medicine, Benha university.

## 2.2. Study design

The study was conducted from January to December 2023 across four governorates in the Nile Delta region of Egypt: Gharbia, Kafr El-Sheikh, Menofia, and Qalyubia, geographically situated at 30.867°N 31.028°E, 31°06′42″N 30°56′45″E, 0.52°N 30.99°E and 30.41°N 31.21°E, respectively. Gharbia lies in the central Delta, bordered by Kafr El-Sheikh to the north and Menofia to the south. Kafr El-Sheikh is located in the northernmost part of the Delta, adjacent to the Mediterranean Sea. Menofia is positioned to the south of Gharbia and west of the Nile River, while Qalyubia lies just north of Cairo, on the eastern edge of the Delta. These governorates are characterized by a temperate climate with mild winters and warm, humid summers, creating favorable conditions for agricultural and livestock activities. The Nile Delta is known for its fertile soil and dense agricultural activity, making it one of the most important farming regions in Egypt. These areas have a high population of domestic animals, particularly buffaloes, which play a vital role in local dairy and meat production. The close interaction between animals and humans, combined with traditional farming practices, may contribute to the transmission of zoonotic diseases, emphasizing the importance of conducting epidemiological studies in such settings.

## 2.3. Sample size and sampling

A cross-sectional study was conducted during the period from January to December 2023 and the samples were selected through simple random sampling from herds raised in four governorates under investigation. The sample size was determined based on the formula described by Thrusfield [25]:

$$n = (1.96)^2 \times P(1 - P)/ d^2$$

where $n$ is the required sample size, *1.96* is the Z-value corresponding to a 95% confidence level, $P$ represents the expected prevalence (12%, as reported by Mo'awad, Sobhy [24]), and $d$ is the desired precision (5%). Based on this calculation, a total of 440 blood samples were collected from randomly selected cattle. Blood samples (5 ml each) were collected from jugular vein of each examined water buffaloes using Blain vacutainers. The samples were transported in an ice box to the Veterinary Diagnostic Laboratory, Faculty of Veterinary Medicine, Benha University. Serum was separated by centrifugation at 4000 × g for 5 minutes and stored at −20°C until further serological analysis.

During sampling, farmers were provided with a structured questionnaire designed to collect data on potential risk factors associated with *C. burnetii* infection. The questionnaire included variables such as geographical location, sex of the animal (male or female), age categories (2–5 years, > 5–8 years, and >8 years), communal grazing practices, contact with small ruminants, presence of tick infestation, and history of abortion.

## 2.4. Serological analysis

Serum samples were tested for antibodies against *C. burnetii* using a commercial indirect ELISA kit (IDEXX Q-Fever, IDEXX Laboratories, USA). The test was conducted in full compliance with the manufacturer's protocol, without any alterations. Briefly, serum samples along with positive and negative controls were diluted 1:400 using 1 × wash solution. Then, 100 µL of each diluted sample and control was added to the appropriate wells of the antigen-coated microtiter plate. The plate was incubated at 37ºC for 60 min. After washing the wells three times, 100 µL of conjugate was added to each well, followed by a second incubation at 37ºC for 60 min. Following another three washes, TMB substrate was added, and the plate was incubated at 18–26ºC for 15 min. Finally, the enzymatic reaction was stopped by adding a stop solution.

Optical density (OD) readings were measured at 450 nm using an ELISA reader (AMR-100, AllSheng, China). The sample-to-positive percentage (S/P%) for each tested sample was calculated following the manufacturer's instructions. According to these guidelines, samples with S/P% values above 40 were considered positive, those below 30 were negative, and values between 30 and 40 were classified as suspect.

 

The S/P% was calculated using the formula:

$$S/P\% = 100 \times \text{Sample A}(450) - NC/PC - NC$$

where *Sample* A450 is the optical density of the sample, PC is the mean optical density of the positive control, and NC is the mean optical density of the negative control.

## 2.5. Statistical analysis

The data obtained from the serological survey were analyzed using SPSS software version 24 (IBM, USA). Univariable analysis was performed using the chi-square test for each variable. Variables with a *P*-value less than 0.2 were included in the multivariable logistic regression analysis. A backward stepwise selection approach was applied, using the likelihood ratio test at each step, with a significance threshold of 0.05 for entry and 0.1 for removal. Variables with *P* < 0.05 were considered statistically significant. The Hosmer–Lemeshow goodness-of-fit test was used to assess the overall fit of the final regression model [26].

## 3. Results

Based on univariable analysis, the seroprevalence of *C. burnetii* in buffaloes did not differ significantly across the studied governorates (*P* > 0.05). The highest seroprevalence was recorded in Kafr El-Sheikh (15.5%), while the lowest was observed in Menofia (9.1%). Similarly, sex was not found to have a significant effect on seroprevalence; however, female buffaloes exhibited a higher prevalence rate (12.4%) compared to males (9%), Table 1.

Age was found to have a significant influence on the seroprevalence of *C. burnetii* among the examined buffaloes, with older animals showing higher infection rates. Buffaloes over 8 years of age had the highest seroprevalence (21.6%), compared to only 5% in those aged 2–5 years. Additionally, the presence of ticks was significantly associated with increased seroprevalence, reaching 22.4% in tick-infested animals. Contact with other animals, either through communal grazing or exposure to small ruminants, also had a notable impact. Buffaloes involved in communal grazing had a seroprevalence of 13.8%, while those in contact with small ruminants showed a prevalence of 16.3% (Table 1).

Interestingly, the history of abortion was strongly associated with a higher seroprevalence of *C. burnetii*. Animals with a history of abortion exhibited a prevalence rate of 26.7%, significantly higher than that of non-aborted animals, Table 1.

Multivariate logistic regression analysis revealed that buffaloes older than 8 years had significantly higher odds of testing positive for *C. burnetii* (odds ratio = 6.7, 95% CI: 2.5–17.8), followed by those aged between 5 and 8 years (odds ratio = 2.2, 95% CI: 0.8–5.9), when compared to younger animals (2–5 years). The likelihood of infection also increased among buffaloes involved in communal grazing (odds ratio = 1.9, 95% CI: 0.9–4.2) and those infested with ticks (odds ratio = 3.0, 95% CI: 1.5–5.9). Moreover, animals in contact with small ruminants (odds ratio = 3.0, 95% CI: 1.5–6.2) and those with a history of abortion (odds ratio = 3.2, 95% CI: 1.4–7.4) were significantly more likely to be seropositive compared to other animals, Table 2.

## 4. Discussion

In Egypt, *C. burnetii* is frequently detected in sheep, goats, and cattle herds, and is known to play a significant role in causing abortions in sheep, resulting in notable economic losses to the livestock industry. Despite this, there is a lack of sufficient information regarding the prevalence and impact of Q fever in other ruminants, especially water buffaloes.

To better understand the current status of *C. burnetii* infection among water buffaloes in Egypt's Nile Delta, we investigated the seroprevalence of *C. burnetii* in water buffaloes raised in this region and identified the associated risk factors.

The overall seroprevalence of *C. burnetii* in water buffaloes examined across the studied governorates in Egypt's Nile Delta was 11.6%, aligning closely with a previously reported rate of 12% in selected governorates of Upper Egypt [24].

**Table 1.** Univaraible analysis of *C. burnetii* seroprevalence in buffaloes across the studied locations in relation to various risk factors.

| Variable | No of tested samples | No of positive | No of Negative | % | 95%Confidence interval | P value |
|---|---|---|---|---|---|---|
| **Location** | | | | | | |
| Gharbia | 110 | 11 | 99 | 10.0 | 5.68-17.02 | χ2 = 2.551 df = 3 P = 0.446 |
| Kafr ElSheikh | 110 | 17 | 93 | 15.5 | 9.88-15.45 | |
| Menofia | 110 | 10 | 100 | 9.1 | 5.01-15.93 | |
| Qalyubia | 110 | 13 | 97 | 11.8 | 7.04-19.18 | |
| **Sex** | | | | | | |
| Male | 100 | 9 | 91 | 9.0 | 4.81-16.23 | χ2 = 0.848 df = 1 P = 0.357 |
| Female | 340 | 42 | 298 | 12.4 | 9.27-16.27 | |
| **Age** | | | | | | |
| 2-5 | 120 | 6 | 114 | 5.0 | 2.31-10.48 | χ2 = 18.367 df = 2 P < 0.0001* |
| 5-8 | 195 | 18 | 177 | 9.2 | 5.92-14.12 | |
| >8 | 125 | 27 | 98 | 21.6 | 15.29-29.6 | |
| **Communal grazing area** | | | | | | |
| No | 135 | 9 | 126 | 6.7 | 3.55-12.19 | χ2 = 4.608 df = 1 P = 0.032* |
| Yes | 305 | 42 | 263 | 13.8 | 10.35-18.09 | |
| **Tick infestation** | | | | | | |
| Yes | 85 | 19 | 66 | 22.4 | 14.8-32.29 | χ2 = 11.906 df = 1 P = 0.001* |
| No | 355 | 32 | 323 | 9.0 | 6.45-12.44 | |
| **Presence of small ruminants** | | | | | | |
| No | 200 | 12 | 188 | 6.0 | 3.47-10.19 | χ2 = 11.185 df = 1 P = 0.001* |
| Yes | 240 | 39 | 201 | 16.3 | 12.12-21.44 | |
| **History of abortion in previous year** | | | | | | |
| No | 395 | 39 | 356 | 9.9 | 7.3-13.21 | χ2 = 11.118 df = 1 P = 0.001* |
| Yes | 45 | 12 | 33 | 26.7 | 15.97-41.04 | |
| Total | 440 | 51 | 389 | 11.6 | 8.93-14.92 | |

*Results were considered statistically significant when the P-value was less than 0.05.

Seroprevalence rates of *C. burnetii* among bovine populations in various European countries have ranged from 6% to 15% in previous studies. For instance, reported rates were 7.8% in Germany, 6.7% in Spain, and 7.9% in Albania [27–29]. More recently, a serological survey in Poland recorded a prevalence of 4.18% [30]. Collectively, these studies demonstrate that *C. burnetii* is widely distributed in European bovine populations. However, variations in prevalence may be attributed to differences in regional epidemiological conditions, sample sizes, and the specific ELISA kits employed in testing [31,32].

The epidemiological profile of *C. burnetii* in buffalo appears to differ somewhat from that in cattle. To date, only a limited number of studies have investigated the seroprevalence of *C. burnetii* in buffaloes. Countries such as India, Egypt, and Thailand have conducted large-scale serological surveys in this species, with reported prevalence rates ranging from 5% to 20% [33–39].

Including other susceptible animal species such as goats, in addition to cattle as previously studied, would provide a broader epidemiological perspective on *C. burnetii* infection across ruminant populations. Goats are known to be highly susceptible to *C. burnetii* and are often implicated in zoonotic outbreaks, particularly due to their close contact with humans and tendency to shed large quantities of the pathogen during parturition. Several studies in Egypt have reported seroprevalence rates in goats ranging from 13% to over 30%, which are often higher than those observed in cattle or buffaloes [21,33,40,41].

**Table 2. Multivariable logistic regression analysis for *C. burnetii* regressed against four variables as risk factors from N=(440) buffaloes in four governorates in Egypt's Nile Delta.**

| Variable | B | S.E. | OR | 95% C.I. for OR | | P value |
|---|---|---|---|---|---|---|
| | | | | Lower | Upper | |
| **Age** | | | | | | |
| 2-5 | reference | | | | | |
| 5-8 | 0.782 | 0.507 | 2.2 | 0.8 | 5.9 | 0.012 |
| >8 | 1.903 | 0.499 | 6.7 | 2.5 | 17.8 | <0.0001 |
| **Communal grazing area** | | | | | | |
| No | reference | | | | | |
| Yes | 0.639 | 0.405 | 1.9 | 0.9 | 4.2 | 0.011 |
| **Tick infestation** | | | | | | |
| No | reference | | | | | |
| Yes | 1.089 | 0.354 | 3.0 | 1.5 | 5.9 | 0.002 |
| **Presence of small ruminants** | | | | | | |
| No | reference | | | | | |
| Yes | 1.111 | 0.363 | 3.0 | 1.5 | 6.2 | 0.002 |
| **History of abortion in previous year** | | | | | | |
| No | reference | | | | | |
| Yes | 1.175 | 0.423 | 3.2 | 1.4 | 7.4 | 0.005 |

B: Logistic regression coefficient, SE: Standard error, OR: Odds ratio, 95%CI: Confidence interval.

The variation in *C. burnetii* seroprevalence among buffalo populations across countries is influenced by multiple factors. These include differences in regional epidemiological conditions such as animal density, health status, and vector activity, as well as variations in animal husbandry practices like housing and grazing systems. Environmental factors, including climate and geography, also play a role in disease transmission. Additionally, geographical factors such as shared grazing areas and dense animal populations in certain regions may increase the likelihood of contact between infected and susceptible animals, thereby facilitating disease spread. Moreover, discrepancies in diagnostic tools, testing protocols, sample sizes, and study designs can lead to differing results [8,10,13,19,42–44].

Among all the tested animal sera, females showed a higher seropositivity rate (12.4%) compared to males (9%). These results align with previous studies that reported a greater prevalence in females than in males [45]. The increased seroactivity in females could be attributed to pregnant buffaloes' heightened susceptibility and the organism's continuous shedding into the environment following normal parturition or abortion via milk, amniotic foetal membranes, fluid and vaginal discharge [46].

In the present study, we observed that the likelihood of testing positive for *C. burnetii* increased with age, a finding consistent with previous studies [45]. Similar age-related trends in seropositivity were noted by Mazeri, Scolamacchia [47] in Cameroon and Keshavamurthy, Singh [48] in India. This pattern is likely attributed to the cumulative risk of exposure to the pathogen over time [3].

Notably, a significant association (P<0.01) was found between *C. burnetii* seropositivity and contact with other herds through shared grazing areas, consistent with the observations of Adamu, Kabir [49]. *C. burnetii* is known for its strong environmental resilience, remaining infectious for months under favorable conditions [1]. Additionally, vaginal discharges, birth products, urine, feces, and milk from infected animals significantly contribute to environmental contamination and facilitate the spread of infection among animals during communal grazing [50].

There was a statistically significant correlation (P=0.001) between tick infestations and the seropositivity for *C. burnetii-specific* infection. These findings are similar with prior research by Rashid, Saqib [45], who found that ticks play

a key role in the transmission and duration of the disease in both animals and people. These findings confirmed the role of ticks in the transmission of *C. burnetii* in buffaloes. The bacteria can multiply within ticks and be transmitted through their bites or by contamination of the environment with tick feces or crushed tick bodies. Infected ticks feeding on multiple animals help spread the infection within and between herds [13,51–53].

Moreover, it was found that the seroprevalence of *C. burnetii* antibodies in buffaloes significantly increased when they were raised in close contact with small ruminants. This observation is consistent with the findings of Menadi, Mura [54] and Selim, Marawan [3], who reported a higher infection risk and greater genetic diversity of *C. burnetii* in mixed-species herds. The increased seropositivity can be explained by the fact that small ruminants, particularly goats and sheep, are well-known reservoirs and prolific shedders of *C. burnetii*, especially during parturition. Their close interaction with buffaloes in shared housing or grazing areas facilitates cross-species transmission, contributing to higher exposure and infection rates in buffaloes [4,23,33,55].

The present findings indicate that buffaloes with a history of abortion exhibited higher seropositivity for *C. burnetii* compared to those without such reproductive issues. This observation aligns with the results reported by Menadi, Mura [54]. The increased seropositivity in these animals may be attributed to the greater susceptibility of pregnant ruminants to *C. burnetii* infection. The bacterium has a particular affinity for the reproductive tract and is known to cause reproductive disorders, including abortion and infertility [56]. Infected gravid animals may experience placental colonization, leading to pregnancy loss and the shedding of large numbers of bacteria during abortion or parturition, thereby contributing to both individual infection and environmental contamination [8,57].

One potential limitation of this study is the use of ELISA for detecting *C. burnetii*-specific antibodies. While ELISA is a practical and widely used serological method, the presence of antibodies does not necessarily indicate current infection, as it reflects prior exposure rather than active shedding of the pathogen. Additionally, the sensitivity and specificity of ELISA may be lower than that of molecular techniques such as PCR, which could lead to minor under- or overestimation of the true prevalence [58–61].

## Conclusion

This study highlights the epidemiological importance of *C. burnetii* infection in buffaloes in Egypt's Nile Delta, revealing a seroprevalence rate of 11.6%, similar to rates in other regions and countries. The infection was significantly associated with factors such as female sex, older age, tick infestation, shared grazing, contact with small ruminants, and a history of abortion. These findings suggest that *C. burnetii* transmission in buffaloes is influenced by environmental persistence and interactions with other animals. The increased risk in reproductively compromised animals emphasizes the bacterium's role in economic losses. Therefore, enhanced surveillance and biosecurity measures are recommended to control Q fever in buffalo populations.

## Author contributions

**Conceptualization:** Abdelfattah Selim, Hattan S. Gattan, Hesham Ismail.

**Data curation:** Abdelfattah Selim, Mohamed Marzok, Hattan S. Gattan.

**Formal analysis:** Abdelfattah Selim, Mohamed Marzok, Hesham Ismail.

**Funding acquisition:** Abdelfattah Selim, Hesham Ismail.

**Investigation:** Hattan S. Gattan.

**Methodology:** Abdelfattah Selim, Mohamed Marzok, Hattan S. Gattan, Hesham Ismail.

**Software:** Abdelfattah Selim, Hesham Ismail.

**Supervision:** Abdelfattah Selim, Mohamed Marzok.

**Validation:** Hattan S. Gattan.

**Visualization:** Mohamed Marzok, Hattan S. Gattan, Hesham Ismail.

**Writing – original draft:** Abdelfattah Selim, Mohamed Marzok, Hattan S. Gattan, Hesham Ismail.

**Writing – review & editing:** Abdelfattah Selim.

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
