## [Decision Letter · Decision Letter 0]

2 Aug 2025

Dear Dr.  Selim,

Thank you for submitting your manuscript to PLOS ONE. After careful consideration, we feel that it has merit but does not fully meet PLOS ONE’s publication criteria as it currently stands. Therefore, we invite you to submit a revised version of the manuscript that addresses the points raised during the review process.

**ACADEMIC EDITOR:**
**Please, revise the manuscript accordingly to the reviewers comments (including the comments provided by myself)**

We look forward to receiving your revised manuscript.

Kind regards,

Gianmarco Ferrara

Academic Editor

PLOS ONE

Journal Requirements:

2. To comply with PLOS ONE submissions requirements, in your Methods section, please provide additional information regarding the experiments involving animals and ensure you have included details on methods of anesthesia and/or analgesia

4. In the online submission form, you indicated that “available from corresponding author on request”

All PLOS journals now require all data underlying the findings described in their manuscript to be freely available to other researchers, either;

1. In a public repository

2. Within the manuscript itself

3. Uploaded as supplementary information.

Additional Editor Comments:

One reviewer have problems in the loading the report. Their comments are as follows:

General comment: water buffalo are an integral part of Egyptian livestock

Did the authors investigate the possible effects of climate change on the seroprevalence and

transmission dynamics?

Additional knowledge gained from including other susceptible animals (goats, as was done for

cattle) or at least a statement of the seroprevalence of the problem in these species would have

added value to the results and a comparative aspect of the problem across species.

Good and well written article which should qualify for publication as a short communication. It

requires more work to qualify as a peer reviewed journal article. Eg

• comparison of buffalo and cattle and goat seroprevalence

• indications and elaboration on routes of transmission and extent of problem in

humans

• the effect of climate change on the transmission dynamics of the disease and

preparedness steps that can be taken.

Line

71 Benha University and every where else where this appears

78 Study design must be bold

This qualifies as a short communication

Reviewers' comments:

Reviewer's Responses to Questions

**Comments to the Author**

1. Is the manuscript technically sound, and do the data support the conclusions?

Reviewer #1: Yes

Reviewer #2: Partly

Reviewer #3: Yes

2. Has the statistical analysis been performed appropriately and rigorously?

Reviewer #1: Yes

Reviewer #2: N/A

Reviewer #3: No

3. Have the authors made all data underlying the findings in their manuscript fully available?

Reviewer #1: Yes

Reviewer #2: No

Reviewer #3: No

4. Is the manuscript presented in an intelligible fashion and written in standard English?

Reviewer #1: Yes

Reviewer #2: Yes

Reviewer #3: Yes

Reviewer #1: The manuscript entitled “Risk Factors Influencing Coxiella burnetii Seropositivity in Buffalo Populations of Egypt’s Nile Delta” is clear and well written. The methodology is adequate. The discussion is clear and objective. Below are minor comments for consideration.

Comments:

Line 99: Please consider including the blood collection volume, tube type, and site (ie. Tail vein, jugular vein, etc..).

Lines 93 and 100: This states that cattle were sampled and blood was collected from cattle, respectively. Shouldn’t this be from buffalo?

Line 104: It may be helpful to have the questionnaire available for to the reader, if possible.

Line 108: Please include a statement and/or include references that clarifies to the reader that the IDEXX kit works for buffalo serum. The test has been used with buffalo serum in other studies, but the extra clarity for the reader would be useful since the manufacturer specifies its use in sheep, goats, and cattle.

Line 127: Please include how confounders were handled.

Lines 138 – 151: Please include clarity throughout this section so the reader knows this is based on univariable analysis. Similarly, please clarify in the title of Table 1 as well.

Lines 190-195: Suggest removing this section since the findings were not at the significance threshold set by the authors.

Lines 193 and 229: Please include a reference to support the statements that pregnant females have greater susceptibility to C. burnetii infection.

Line 238: The statement that female sex is significantly associated with infection does not align with the results section or Tables. Additionally, infection status of the animals is not assessed in this study, rather seropositivity.

Table 1:

- Suggest removing “locality” from the first row.

- The > before 5-8 should be removed. Same for Table 2. Same for line 106.

- How is tick infestation defined? Please clarify somewhere in the manuscript. Is it the presence of any ticks or > a certain number? Is the status determined at the time of blood collection?

- yes in the “History of abortion in previous year” row is lowercase.

- The inconsistency between the position of Yes and No in the rows for Tick infestation could lead to confusion since it is different from the other variables.

Reviewer #2: The authors have tried to summarize risk factors that can cause or influence Coxiella burnetii positive in buffalo serum. Though, there are some points that should be included or added in the manuscript.

Major points:

- The authors detected Coxiella burnetii antibodies in buffalo serum samples using a commercial indirect ELISA. It would be nice if they can include or at least perform PCR reaction in DNA extraction of blood sample (additional from serum) and sequence the samples to confirm the sequence of DNA that showed Coxiella bunetii positive. No need to sequence all the samples but at least some to confirm the results.

- The Coxiella bunetii DNA sequence can provide additional valuable information and analyze further to increase the impact of this research/manuscript.

- Even though, the Ethics approval is mentioned, the authors should add the “protocol number” of this research, especially this manuscript is dealing with buffaloes which are mammals.

-There are still more factors that should be considered, such as, the way the buffalo live, antibiotic treated and etc. Though, the authors did not measure these factors but they should discuss or include the background of these factors in the population being studied.

Minor points:

- Please keep the consistency of how to write the word Coxiella burnetii, “full name” or “abbreviated” when mentioned for the first time and later.

- Add full word of “OR” in text

- Add GPS for each location being studied

- Add genus of the buffalo

- Table 1 heading, write the full word of “CI”

- How big of the number of tick sample for tick infestation?

- What the authors mean by presence of small ruminants?

Reviewer #3: The manuscript presents an important investigation on C. burnetii in water buffalo, which has an important public health significance; however, several improvements in structure, clarity, and language are necessary to enhance its scientific value and reproducibility.

Abstract

• Line 23-24: Please rewrite “Buffaloes over 8 years of age showed a higher prevalence (21.6%) compared to the younger age groups”

Materials and methods

• Line 93: Authors quoted “conducted using a convenient sample of cattle, selected through simple random sampling from herds…”. Did they mean a convenient sample of the herds and a random sampling of animals within herds? Please explain how this was actually done to make the study reproducible.

• Line 99-100: A more details of sample collection is required, volume of each sample collected in which type of container (e.g., EDTA or clot activator etc.), who collected the samples, was the samples taken in ice box immediately, and a time range within which the samples were processed in lab for centrifugation, and how long the samples were stored in -20°C before serological analysis?

• Line 128-134: The Authors did not address confounding, interaction, and collinearity for the final regression model. If any confounding is found that variable must be included in the final model to correct for the confounding effect. If there is collinearity, only one variable should be chosen to eliminate the duplication of effects.

• For me, this is actually a multivariable regression, not multivariate, since the outcome is binary (yes/no)

Results

• Line 413 Table 2: Please keep the reference category for each variable in the table to make the comparison easily understandable. Please improve the table captions e.g., Multivariable logistic regression analysis for C. burnetii regressed against four variables as risk factors from N=(440) buffaloes in four governorates in Egypt’s Nile Delta

• Line 415: CI: 95% Confidence intervals

• Results of confounding and collinearity check should be mentioned in results

Discussion

• My suggestions are that the discussion may start with a brief paragraph on the important findings, especially the higher prevalence and risk factors how it could be important for the perspective of the study population. This partly done in line 165-167 where author can specify the risk factors e.g., older buffaloes, communal grazing area, presence of tick infestation, small ruminants, and previous history of abortion.

• Line 171 “Seroprevalence rate” rate should be carefully interpreted, rate usually is relevant to incidence, which is different from prevalence.

• Could the ELISA be a limitation for example, antibody may not always represent actual presence of the pathogen and since the Se and Sp of ELISA might be lower than PCR? The authors may also report this, which could result in a minor underestimation or overestimation of actual prevalence.

• Line 184-188: Authors may explain the source of variations e.g., “Environmental factors, including climate and geography, also play a role in disease transmission.” How the transmission occurs in response to 1 or 2 of those important variables.

• “discrepancies in diagnostic tools, testing protocols, sample sizes, and study designs can lead to differing results ” Similarly here, authors may explain here 1 or 2 of those important variables. That will make the discussion more informative.

• Line 204-207 “Additionally, vaginal discharges, …… animals during communal grazing [43].” Does it mean that in communal grazing, the management seems more difficult than in non-grazing?

**Do you want your identity to be public for this peer review?** For information about this choice, including consent withdrawal, please see our Privacy Policy

Reviewer #1: No

Reviewer #2: No

Reviewer #3: **Yes: ** Shuvo Singha

---

## [Author Response · Author response to Decision Letter 1]

5 Aug 2025

General comment:

water buffalo are an integral part of Egyptian livestock

Point#: Did the authors investigate the possible effects of climate change on the seroprevalence and transmission dynamics?

Response#: We thank the reviewer for this insightful comment. While our study did not directly assess the impact of climate change on the seroprevalence and transmission dynamics of Coxiella burnetii, we acknowledge that changing climatic conditions—such as increased temperature, altered rainfall patterns, and more frequent dust storms—may influence the survival and dissemination of the pathogen in the environment. These factors could potentially affect exposure risk in both animals and humans.

Point#: Additional knowledge gained from including other susceptible animals (goats, as was done for cattle) or at least a statement of the seroprevalence of the problem in these species would have added value to the results and a comparative aspect of the problem across species.

Response#: it added

Point#: 71 Benha University and every where else where this appears

Response#: it added

Point#: 78 Study design must be bold

Response#: it added

Reviewer 2

The manuscript presents an important investigation on C. burnetii in water buffalo which has an important public health significance; however, several improvements in structure, clarity, and language are necessary to enhance its scientific value and reproducibility.

Abstract

Point#: Line 23-24: Please rewrite “Buffaloes over 8 years of age showed a higher prevalence (21.6%) compared to the younger age groups”

Response#: It was rewritten

Materials and methods

Point#: Line 93: Authors quoted “conducted using a convenient sample of cattle, selected through simple random sampling from herds…”. Did they mean a convenient sample of the herds and a random sampling of animals within herds?

Response#: It was edited

Point#: Line 99-100: A more details of sample collection is required, volume of each sample collected in which type of container (e.g., EDTA or clot activator etc.), who collected the samples, was the samples taken in ice box immediately, and a time range within which the samples were processed in lab for centrifugation, and how long the samples were stored in -20°C before serological analysis?

Response#: It was clarified

Point#: For me, this is actually a multivariable regression, not multivariate, since the outcome is binary (yes/no)

Response#: It was edited

Results

Point#: Line 413 Table 2: Please keep the reference category for each variable in the table to make the comparison easily understandable. Please improve the table captions e.g., Multivariable logistic regression analysis for C. burnetii regressed against four variables as risk factors from N=(440) buffaloes in four governorates in Egypt’s Nile Delta

Response#: It was edited

Point#: Line 415: CI: 95% Confidence intervals

Response#: It was edited

Discussion

Point#: My suggestions are that the discussion may start with a brief paragraph on the important findings, especially the higher prevalence and risk factors how it could be important for the perspective of the study population. This partly done in line 165-167 where author can specify the risk factors e.g., older buffaloes, communal grazing area, presence of tick infestation, small ruminants, and previous history of abortion.

Response#: It was edited

Point#: Could the ELISA be a limitation for example, antibody may not always represent actual presence of the pathogen and since the Se and Sp of ELISA might be lower than PCR? The authors may also report this, which could result in a minor underestimation or overestimation of actual prevalence.

Response#: It was added

Point#: Line 184-188: Authors may explain the source of variations e.g., “Environmental factors, including climate and geography, also play a role in disease transmission.” How the transmission occurs in response to 1 or 2 of those important variables.

Response#: It explained

Point#: “discrepancies in diagnostic tools, testing protocols, sample sizes, and study designs can lead to differing results ” Similarly here, authors may explain here 1 or 2 of those important variables. That will make the discussion more informative.

Response#: It explained

Point#: Line 204-207 “Additionally, vaginal discharges, …… animals during communal grazing [43].” Does it mean that in communal grazing, the management seems more difficult than in non-grazing?

Response#: It facilitate direct contact with source of infection

Reviewers' comments:

Reviewer's Responses to Questions

Comments to the Author

1. Is the manuscript technically sound, and do the data support the conclusions?

Response#: it edited

2. Has the statistical analysis been performed appropriately and rigorously?

Response#: it edited

3. Have the authors made all data underlying the findings in their manuscript fully available?

Response#: it edited

4. Is the manuscript presented in an intelligible fashion and written in standard English?

Response#: it edited

5. Review Comments to the Author

Reviewer #1:

Comments:

Line 99: Please consider including the blood collection volume, tube type, and site (ie. Tail vein, jugular vein, etc..).

Response#: it edited

Lines 93 and 100: This states that cattle were sampled and blood was collected from cattle, respectively. Shouldn’t this be from buffalo?

Response#: it edited

Line 104: It may be helpful to have the questionnaire available for to the reader, if possible.

Response#: it edited

Lines 193 and 229: Please include a reference to support the statements that pregnant females have greater susceptibility to C. burnetii infection.

Response#: it edited

Table 1:

- Suggest removing “locality” from the first row.

Response#: it edited

Reviewer #2:

The authors have tried to summarize risk factors that can cause or influence Coxiella burnetii positive in buffalo serum. Though, there are some points that should be included or added in the manuscript.

Major points:

- The authors detected Coxiella burnetii antibodies in buffalo serum samples using a commercial indirect ELISA. It would be nice if they can include or at least perform PCR reaction in DNA extraction of blood sample (additional from serum) and sequence the samples to confirm the sequence of DNA that showed Coxiella bunetii positive. No need to sequence all the samples but at least some to confirm the results.

- The Coxiella bunetii DNA sequence can provide additional valuable information and analyze further to increase the impact of this research/manuscript.

Response#: the manuscript focused on Sero epidemiology and assessment of associated risk factors

- Even though, the Ethics approval is mentioned, the authors should add the “protocol number” of this research, especially this manuscript is dealing with buffaloes which are mammals.

Response#: it added

-There are still more factors that should be considered, such as, the way the buffalo live, antibiotic treated and etc. Though, the authors did not measure these factors but they should discuss or include the background of these factors in the population being studied.

Response#: we studied the available factors

Minor points:

- Please keep the consistency of how to write the word Coxiella burnetii, “full name” or “abbreviated” when mentioned for the first time and later.

Response#: it edited

- Add full word of “OR” in text

Response#: it edited

- Add GPS for each location being studied

Response#: it added

- Add genus of the buffalo

Response#: it added

- Table 1 heading, write the full word of “CI”

Response#: it edited

- What the authors mean by presence of small ruminants?

Response#: it contact of examined buffalo by goat and sheep

---

## [Decision Letter · Decision Letter 1]

27 Aug 2025

Dear Dr. Selim,

Thank you for submitting your manuscript to PLOS ONE. After careful consideration, we feel that it has merit but does not fully meet PLOS ONE’s publication criteria as it currently stands. Therefore, we invite you to submit a revised version of the manuscript that addresses the points raised during the review process.

We look forward to receiving your revised manuscript.

Kind regards,

Gianmarco Ferrara

Academic Editor

PLOS ONE

Journal Requirements:

Additional Editor Comments:

Some previous comments have been not addressed.

Reviewers' comments:

Reviewer's Responses to Questions

**Comments to the Author**

Reviewer #1: (No Response)

Reviewer #3: All comments have been addressed

2. Is the manuscript technically sound, and do the data support the conclusions?

Reviewer #1: Partly

Reviewer #3: Yes

3. Has the statistical analysis been performed appropriately and rigorously?

Reviewer #1: I Don't Know

Reviewer #3: Yes

4. Have the authors made all data underlying the findings in their manuscript fully available?

Reviewer #1: No

Reviewer #3: Yes

5. Is the manuscript presented in an intelligible fashion and written in standard English?

Reviewer #1: Yes

Reviewer #3: Yes

Reviewer #1: A number of comments from the previous round were not addressed. Please see them below.

Lines 93 and 100: This states that cattle were sampled and blood was collected from cattle, respectively. Shouldn’t this be from buffalo?

- Line 101 states cattle were sampled.

Line 127: Please include how confounders were handled.

- No response was provided.

Lines 138 – 151: Please include clarity throughout this section so the reader knows this is based on univariable analysis. Similarly, please clarify in the title of Table 1 as well.

- No response was provided.

Lines 190-195: Suggest removing this section since the findings were not at the significance threshold set by the authors.

- No response was provided.

Lines 193 and 229: Please include a reference to support the statements that pregnant females have greater susceptibility to C. burnetii infection. Not addressed

- No response was provided.

Line 238: The statement that female sex is significantly associated with infection does not align with the results section or Tables. Additionally, infection status of the animals is not assessed in this study, rather seropositivity.

- No response was provided.

Table 1:

- The > before 5-8 should be removed. Same for Table 2. Same for line 106.

- How is tick infestation defined? Please clarify somewhere in the manuscript. Is it the presence of any ticks or > a certain number? Is the status determined at the time of blood collection?

- yes in the “History of abortion in previous year” row is lowercase.

New Comments

- Line 101: please clarify what pf stands for.

Reviewer #3: No further comments on the updated version. Thanks to the authors for sufficiently revising the manuscript.

**Do you want your identity to be public for this peer review?** For information about this choice, including consent withdrawal, please see our Privacy Policy

Reviewer #1: No

Reviewer #3: **Yes: ** Shuvo Singha

---

## [Author Response · Author response to Decision Letter 2]

28 Aug 2025

Comments to the Author

1. If the authors have adequately addressed your comments raised in a previous round of review and you feel that this manuscript is now acceptable for publication, you may indicate that here to bypass the “Comments to the Author” section, enter your conflict of interest statement in the “Confidential to Editor” section, and submit your "Accept" recommendation.

Reviewer #1: (No Response)

Reviewer #3: All comments have been addressed

2. Is the manuscript technically sound, and do the data support the conclusions?

Reviewer #1: Partly

Reviewer #3: Yes

3. Has the statistical analysis been performed appropriately and rigorously?

Reviewer #1: I Don't Know

Reviewer #3: Yes

4. Have the authors made all data underlying the findings in their manuscript fully available?

Reviewer #1: No

Reviewer #3: Yes

5. Is the manuscript presented in an intelligible fashion and written in standard English?

Reviewer #1: Yes

Reviewer #3: Yes

Review Comments to the Author

Reviewer #1: A number of comments from the previous round were not addressed. Please see them below.

Point#: Lines 93 and 100: This states that cattle were sampled and blood was collected from cattle, respectively. Shouldn’t this be from buffalo?

Response#: it corrected

Point#: Line 101 states cattle were sampled.

Response#: it corrected

Point#: Line 127: Please include how confounders were handled.

As described in statistical analysis, potential confounders such as sex, age, communal grazing practices, contact with small ruminants, presence of tick infestation, and history of abortion were considered. Initially, these variables were examined using univariable analysis. Factors with p < 0.20 in the univariable screening were subsequently included in a multivariable logistic regression model to adjust for possible confounding effects. The final model retained variables that remained significant at p < 0.05.

Point#: Lines 138 – 151: Please include clarity throughout this section so the reader knows this is based on univariable analysis. Similarly, please clarify in the title of Table 1 as well.

Response#: it rephrased

Point#: Lines 190-195: Suggest removing this section since the findings were not at the significance threshold set by the authors.

Response#: it removed

Point#: Lines 193 and 229: Please include a reference to support the statements that pregnant females have greater susceptibility to C. burnetii infection.

Response#: reference 46

Point#: Line 238: The statement that female sex is significantly associated with infection does not align with the results section or Tables.

Response#: line 238 discussed that the seroprevalence increased significantly in animals with history of abortion, not discussed the sex

Point#: Table 1:

- The > before 5-8 should be removed. Same for Table 2. Same for line 106.

Response#: it removed

- How is tick infestation defined? Please clarify somewhere in the manuscript. Is it the presence of any ticks or > a certain number? Is the status determined at the time of blood collection?

Response#: it defined as presence on the coat of animals at time of collection and clarified in materials and methods

- yes in the “History of abortion in previous year” row is lowercase.

Response#: it corrected

Point#: Line 101: please clarify what pf stands for.

Response#: it corrected

Reviewer #3:

No further comments on the updated version. Thanks to the authors for sufficiently revising the manuscript.

Thank you for your valuable comments and your time

---

## [Editor Report · Decision Letter 2]

4 Sep 2025

Dear Dr.  Selim,

Thank you for submitting your manuscript to PLOS ONE. After careful consideration, we feel that it has merit but does not fully meet PLOS ONE’s publication criteria as it currently stands. Therefore, we invite you to submit a revised version of the manuscript that addresses the points raised during the review process.

We look forward to receiving your revised manuscript.

Kind regards,

Gianmarco Ferrara

Academic Editor

PLOS ONE

Journal Requirements:

Additional Editor Comments:

I personally reviewed the latest revisions submitted by the authors (due to the unavailability of reviewer #1). The authors did not submit a "track changes manuscript" (the two uploaded files are identical). Furthermore, some comments were not addressed. I ask the authors to address all reviewer comments and reflect these changes in the manuscript, marking them with the track changes tag.

---

## [Author Response · Author response to Decision Letter 3]

13 Sep 2025

Point#: If the reviewer comments include a recommendation to cite specific previously published works, please review and evaluate these publications to determine whether they are relevant and should be cited. There is no requirement to cite these works unless the editor has indicated otherwise.

Response#: the references were reviewed

Point#: Please review your reference list to ensure that it is complete and correct. If you have cited papers that have been retracted, please include the rationale for doing so in the manuscript text, or remove these references and replace them with relevant current references. Any changes to the reference list should be mentioned in the rebuttal letter that accompanies your revised manuscript. If you need to cite a retracted article, indicate the article’s retracted status in the References list and also include a citation and full reference for the retraction notice.

Response#: the references list was reviewed and corrected

Additional Editor Comments:

Point#: I personally reviewed the latest revisions submitted by the authors (due to the unavailability of reviewer #1). The authors did not submit a "track changes manuscript" (the two uploaded files are identical). Furthermore, some comments were not addressed. I ask the authors to address all reviewer comments and reflect these changes in the manuscript, marking them with the track changes tag.

Response#: comments of reviewer 1 were addressed by red color

---

## [Editor Report · Decision Letter 3]

17 Sep 2025

Risk Factors Influencing Coxiella burnetii Seropositivity in Buffalo Populations of Egypt's Nile Delta

PONE-D-25-32343R3

Dear Dr. Abdelfattah Selim,

We’re pleased to inform you that your manuscript has been judged scientifically suitable for publication and will be formally accepted for publication once it meets all outstanding technical requirements.

Kind regards,

Gianmarco Ferrara

Academic Editor

PLOS ONE
---

## [Editor Report · Acceptance letter]

PONE-D-25-32343R3

PLOS ONE

Dear Dr. Selim,

I'm pleased to inform you that your manuscript has been deemed suitable for publication in PLOS ONE. Congratulations! Your manuscript is now being handed over to our production team.

Kind regards,

on behalf of

Prof. Gianmarco Ferrara

Academic Editor

PLOS ONE